# Coenzyme Q10 and Intracellular Signalling Pathways: Clinical Relevance

**DOI:** 10.3390/ijms262211024

**Published:** 2025-11-14

**Authors:** David Mantle

**Affiliations:** Pharma Nord (UK) Ltd., Morpeth NE61 2DB, Northumberland, UK; dmantle@pharmanord.com

**Keywords:** intracellular signalling pathways, coenzyme Q10, Nrf2/NQO1, Nf-κB, P13/AKT/mTOR, MAPK, JAK/STAT, WNT/B-catenin, AMPK/YAP/OPA1, hedgehog pathway

## Abstract

Intracellular signalling pathways provide a mechanism to connect events at a cell surface to the nucleus and are of fundamental importance to normal cell functioning. Intracellular signalling pathways control many aspects of cell metabolism, including mitochondrial function, oxidative stress, inflammation, and apoptosis/ferroptosis. Randomised controlled clinical trials supplementing coenzyme Q10 (CoQ10) have reported significant clinical improvements in a number of disorders, in turn associated with the action of CoQ10 to promote normal mitochondrial function, reduce oxidative stress and inflammation, and mediate apoptosis and ferroptosis. However, the precise mechanisms by which CoQ10 facilitates beneficial changes in the above factors is not completely understood. In the present article, the evidence we have reviewed provides a supporting rationale that the beneficial role of CoQ10 in the above disorders occurs via mediation of major intracellular signalling pathways, including the Nrf2/NQO1, NF-κB, P13/AKT/mTOR, MAPK, JAK/STAT, WNT/B-catenin, AMPK-YAP-OPA1, and hedgehog (Hh) pathways; the clinical consequences of such mediation are also reviewed.

## 1. Introduction

Intracellular signalling pathways, also known as signal transduction cascades, essentially connect events at the cell surface to the nucleus and are of fundamental importance to normal cell functioning. In a typical pathway, binding of an extracellular molecule to specific receptors on the cell membrane results in the activation of enzymes such as kinases or phosphatases, in turn generating transcription factors that bind to DNA in the nucleus; in this way gene expression is regulated in response to external stimuli [1]. More than 300 individual pathways have been identified in human cells, although these can be broadly classified into less than a dozen general types, depending on the type of cell membrane receptor and mechanism of signal transduction [2]. Examples of external stimuli activating intracellular signalling pathways include hormones and growth factors, such as the binding of epidermal growth factor to its cell surface receptor, triggering the P13/AKT pathway to promote cell proliferation and survival, as detailed in Section 4 of this article.

Coenzyme Q10 (CoQ10) is usually described as a vitamin-like substance, although by definition, CoQ10 is not a vitamin since it is synthesised by most tissues within the human body. CoQ10 has a number of vital functions within cells, particularly in cellular energy production within mitochondria, but also as an antioxidant [3]. Deficiency of CoQ10 has been implicated in the pathogenesis of a number of disorders, and randomised controlled clinical trials supplementing CoQ10 in such disorders have demonstrated significant clinical improvements [4]. These symptomatic improvements are in turn linked with the action of CoQ10 to promote normal mitochondrial function, reduce oxidative stress and inflammation, and mediate apoptosis and ferroptosis [4]. However the precise mechanisms by which CoQ10 facilitates beneficial changes in the above factors are not completely understood. To further elucidate these mechanisms, in the present article we have therefore reviewed evidence for the interaction of CoQ10 with intracellular signalling pathways, since these are known to control many of the factors outlined above, and dysregulation of these pathways has been linked with the pathogenesis of a wide range of disorders [5]. Thus in this article, we have reviewed the potential benefits (for example, in reducing oxidative stress or inflammation) and clinical consequences of supplementary CoQ10 in mediating some of the major intracellular signalling pathways, including the Nrf2/NQO1, NF-κB, P13/AKT/mTOR, MAPK, JAK/STAT, WNT/B-catenin, AMPK-YAP-OPA1, and hedgehog (Hh) pathways.

## 2. The Nrf2/NQO1 Pathway

The Nrf2/NQO1 pathway is a vital defence mechanism, protecting cells from the damaging effects of oxidative stress induced by free radicals, as well as from other types of potentially damaging molecular species. Nuclear factor erythroid 2-related factor 2 (Nrf2) is a transcription factor which acts as a master regulator for cytoprotective gene expression, controlling the production of antioxidant enzymes, detoxification proteins, and other molecules responsible for protecting cells from the effects of harmful substances [6]. The action of Nrf2 is tightly regulated to prevent excessive or insufficient responses to cell stress. Nrf2 is kept inactive in the cytoplasm by a protein called Keap1. When cells encounter oxidative stress (or some other types of stress), Nrf2 is released from Keap1 and translocates to the nucleus, where it binds to specific DNA sequences called antioxidant response elements (AREs), activating the transcription of genes encoding antioxidant and detoxification proteins [7]. In addition, Nrf2 has a role in reducing inflammation by modulating the production of inflammatory molecules and promoting the resolution of inflammatory responses [8]. Nrf2 also plays a role in mitochondrial function, including mitochondrial biogenesis, mitophagy, and autophagy, all of which are essential for maintaining healthy mitochondria [9]. Nrf2 plays a role in stem cell biology, influencing the balance between stem cell self-renewal and differentiation [10].

NAD(P)H quinone dehydrogenase 1 (NQO1) is a Phase II detoxification enzyme that catalyses the two-electron reduction in quinones (and other compounds) to hydroquinones, preventing them from participating in harmful redox cycling, thereby preventing the formation of reactive oxygen free radical species (ROS). NQO1 detoxifies various quinones, including those found in the environment and those produced within the body such as vitamin K. A common polymorphism in the NQO1 gene (C609T) results in a non-functional protein. Individuals homozygous for this polymorphism lack NQO1 activity. NQO1 polymorphisms can affect an individual’s susceptibility to certain diseases and their response to environmental toxins [11].

There are several potential downstream targets of activated Nrf2; in addition to NQO1, these include the enzyme HO1 (haem oxygenase), the products of which have antioxidant, anti-inflammatory, and anti-apoptotic action. Reduced HO-1 levels have been associated with a number of disorders, including miscarriage and pre-eclampsia. Other downstream targets of activated Nrf2 include the antioxidant enzymes glutathione peroxidase, superoxide dismutase, and thioredoxin reductase [12].

The ability of CoQ10 to activate the Nrf2/NQO1 pathway by upregulating the expression of NQO1 and other antioxidant enzymes has implications for various health conditions. For example in diabetes, CoQ10’s ability to activate this pathway and reduce oxidative stress may be beneficial in managing diabetic complications [13]. Additionally, NQO1’s role in stabilizing p53, a tumour suppressor protein, suggests a broader role for CoQ10 in cancer prevention and treatment [14]. In the context of spinal cord injury, CoQ10 has been shown to suppress oxidative stress and apoptosis by activating this pathway, leading to increased levels of Nrf2 and NQO1 and reduced levels of p65, a component of the NF-κB pathway [15]. The above studies represent only a few examples of the action of CoQ10 in activating the Nrf2 pathway; a more complete list of such studies is summarised in Table 1. In every study, CoQ10 administration resulted in increased Nrf2 levels, usually accompanied by increased levels of antioxidant enzymes/reduced levels of oxidative stress, and reduced levels of inflammation and apoptosis.

## 3. The NF-κB Pathway

The nuclear factor kappa-light-chain-enhancer of the activated B cells (NF-κB) pathway is vital for regulating the immune response and inflammation. The NF-κB pathway is activated by diverse stimuli (including stress signals and pathogens) and plays a vital role in both the innate and adaptive immune responses. NF-κB comprises a family of transcription factors that regulate gene expression. The pathway involves the activation of NF-κB dimers, which then translocate to the nucleus to regulate the expression of genes involved in the immune response, inflammation, and apoptosis. NF-κB also has a role in cell proliferation, cell cycle regulation, and neuronal function [38].

CoQ10 can modulate the NF-κB pathway, resulting in reduced inflammation, oxidative stress, and apoptosis. Examples of suppression of the NF-κB pathway by CoQ10 include the reduction in inflammation induced by beta-amyloid in nerve cells [39], and the reduction in oxidative stress and apoptosis following spinal cord injury [15]. A more comprehensive summary of the effects of CoQ10 on the NF-κB pathway is given in Table 2; in every study listed, NF-κB was downregulated, typically accompanied by reductions in oxidative stress, inflammation, or apoptosis. It is of note that NF-κB can regulate the biosynthesis of CoQ10 by binding to specific sites in the COQ7 gene, indicating a feedback loop between CoQ10 and NF-κB [40].

## 4. The PI3K/AKT/mTOR Pathway

The P13K/AKT/mTOR pathway is an intracellular signalling pathway involved in various cell processes, including the promotion of cell survival by inhibiting apoptosis, the stimulation of cell growth and proliferation, and the regulation of glucose metabolism. PI3K refers to phosphoinositide 3-kinases, a family of lipid kinases activated by growth factors and other extracellular signals, which phosphorylate phosphatidylinositol lipids to generate second messengers such as PIP3. AKT refers to protein kinase B, a serine/threonine kinase that plays a central role in downstream signalling that is activated by PIP3 (phosphatidylinositol 3,4,5-trisphosphate). mTOR (mammalian target of rapamycin) is a protein kinase that acts as a downstream effector of AKT, integrating signals from various pathways; it essentially acts as a master regulator of protein synthesis and cell growth [61].

CoQ10 can both activate or inhibit the P13K/AKT/mTOR pathway, depending on the specific cellular context, including cell type or type of cellular stress. For example, in some instances, CoQ10 can promote autophagy by inactivating the PI3K/AKT/mTOR pathway, and in other situations CoQ10 can inhibit autophagy by activating the PI3K/AKT/mTOR pathway [62]; in the latter study, the activation of pancreatic stellate cells was inhibited by suppressing autophagy, leading to anti-fibrotic effects. In other examples, activation of the pathway by CoQ10 has been shown to promote wound healing [63], or to protect against beta-amyloid-induced neuronal death [64]. Inhibition of the pathway by CoQ10 can protect cells from damage caused by heat stress [65] and inhibit the formation of osteoclast cells responsible for bone resorption, thereby preventing osteoporosis [66]. The PI3K/Akt pathway is also involved in stem cell aging, and CoQ10 can inhibit this pathway in mesenchymal stem cells, potentially mitigating cellular aging [67].

## 5. The MAPK Pathway

The MAPK pathway (mitogen-activated protein kinase pathway) is involved in cell growth, differentiation, proliferation, and in stress response. MAPK comprises several related pathways, including the ERK/MAPK pathway activated by growth factors, and the JNK/p38 MAPK pathway activated by stress signals. As with other pathways, binding of extracellular molecular stimuli at specific cell surface receptors results in the activation of the mitogen-activated protein kinase enzymes, which translocate to the nucleus to activate specific genes. Aberrant activation of MAPK pathways is frequently observed in various cancers, and the MAPK pathway is also implicated in inflammatory and neuropathic pain [68].

As with the P13K/AKT/mTOR above, CoQ10 can both activate and inhibit the MAPK pathway. For example, CoQ10 can inhibit the MAPK pathway to reduce inflammation; in rat articular chondrocytes, CoQ10 was found to prevent the interleukin-1 beta-induced inflammatory response by inhibiting the MAPK signalling pathway [69]. CoQ10 also prevented RANKL-induced osteoclastogenesis by promoting autophagy via inactivation of the PI3K/AKT/mTOR and MAPK pathways [70]. In a rat model of spermatogenic dysfunction induced by a high-fat diet, supplementary CoQ10 suppressed testicular dysfunction via inhibition of the MAPK pathway [71]. Alternatively CoQ10 can activate the MAPK pathway to improve memory and neuronal differentiation; in a study investigating the effects of CoQ10 on Alzheimer’s disease in rats, CoQ10 was found to improve memory and neuronal differentiation, partly by activating the TLR-4/MAPK pathway [72]. In chicken myocardial cells, CoQ10 upregulated HSP heat shock protein via the MAPK pathway, demonstrating its role in cellular stress response [73].

## 6. The JAK/STAT Pathway

The JAK-STAT pathway plays a role in the immune response by regulating cytokine signalling, which is essential for inflammation and immune cell development; the pathway is also involved in cell growth, differentiation, and survival. The key components of the pathway are Janus kinases and signal transducers and activators of transcription (STATs). Binding of extracellular molecules (cytokines or growth factors) to specific receptors at the cell surface results in activation of Janus kinases; STAT proteins are in turn activated, translocating to the nucleus, binding to specific DNA regions to activate gene transcription. The pathway is tightly regulated, with feedback mechanisms, such as the suppressor of cytokine signalling proteins, which help to terminate the signal. Dysregulation of the JAK-STAT pathway is implicated in various diseases, including cancers (e.g., leukaemia, lymphoma), autoimmune disorders (e.g., rheumatoid arthritis, inflammatory bowel disease), and other inflammatory conditions [74].

There is evidence that CoQ10 can mediate the JAK/STAT pathway, particularly its role in inflammatory responses. An in silico study identified CoQ10-inducible genes that are functionally connected to the JAK/STAT signalling pathway [75].

## 7. The Wnt/β-Catenin Pathway

The Wnt/β-catenin pathway is a cell-to-cell signalling pathway involved in various biological processes, including embryonic development, tissue homeostasis, cell proliferation, differentiation, and cell polarity. The pathway can be activated through canonical (β-catenin-dependent) or non-canonical (β-catenin-independent) pathways, depending on the cellular context. Wnt refers to extracellular proteins that bind to specific cell surface receptors, in turn activating β-catenin, an intracellular signalling protein that translocates to the nucleus, activating the transcription of target genes involved in cell proliferation, survival, differentiation, and migration.

Dysregulation of the Wnt/β-catenin pathway, often through mutations or epigenetic changes, can lead to uncontrolled cell proliferation and is implicated in various cancers. Aberrant activation of this pathway is also associated with developmental disorders, neurodegenerative diseases, and bone diseases [76].

Research indicates that CoQ10 can play a protective role in neurodegenerative diseases by modulating the Wnt/β-catenin pathway. In a preclinical study examining the effects of cyclophosphamide-induced chemobrain, CoQ10 was found to reduce neuronal apoptosis and preserve hippocampal neurogenesis by influencing the Wnt/β-catenin signalling pathway [77]. In addition, in a preclinical rat model of renal necroinflammation, supplementary CoQ10 attenuated renal fibrosis via mediation of the Wnt/β-catenin pathway [78]. The Wnt/β-catenin pathway plays a fundamental role in cardiac regeneration and tissue repair; this pathway has a key role in promoting cardiomyocyte proliferation during neonatal cardiac development and is re-activated in response to cardiac injury [79]. In this regard the role of supplemental CoQ10 in promoting normal cardiovascular function is of note [80], although direct interaction between CoQ10 and the Wnt/β-catenin signalling pathway in the context of cardiac regeneration has not been reported to date.

## 8. The AMPK-YAP-OPA1 Pathway

The AMPK-YAP-OPA1 pathway is a signalling pathway involved in regulating mitochondrial function, energy metabolism, and cell survival; in effect it provides a mechanism for cells to adapt to changing energy demands. AMPK is an AMP-activated protein kinase, a cellular energy sensor that regulates metabolism that is activated under conditions of cellular stress such as low energy or nutrient deprivation. YAP, or Yes-associated protein, is a transcriptional co-activator involved in cell growth and proliferation. OPA1, or optic atrophy 1, is a protein responsible for mitochondrial fusion and maintaining mitochondrial integrity [81].

A preclinical study in mice provided evidence that CoQ10 can activate the AMPK-YAP-OPA1 pathway to improve mitochondrial function, reduce oxidative stress, and improve energy metabolism, thereby reducing atherosclerosis [82].

## 9. The Hedgehog Pathway

The hedgehog (Hh) pathway is a cell-to-cell communication pathway that plays a key role in organogenesis, stem cell maintenance, and tissue repair, both in embryonic development and in adult tissue homeostasis. So-called hedgehog ligands bind to specific cell surface receptors (PTCH); this in turn activates a protein (SMO), which then activates transcription factors (GLI) in the nucleus. Dysregulation of the Hh pathway is implicated in several diseases, notably cancer, where it can contribute to tumour initiation, progression, and resistance to therapy. Hh inhibitors (HHIs), such as vismodegib and sonidegib, are targeted therapies that block the Hh pathway, helping to slow or stop cancer cell growth [83].

In patients with advanced basal cell carcinoma treated with Hh pathway inhibitors, CoQ10 supplementation can reduce the need for dose reductions due to the side effects of medications such as vismodegib and sonidegib, particularly muscle spasms and fatigue [84].

## 10. Discussion

One might first question why there are so many individual intracellular signalling pathways performing broadly similar functions; however, there is an argument that having multiple pathways that can achieve similar outcomes provides a level of redundancy. If one pathway is compromised, others can compensate, ensuring the cell can still carry out essential functions. In addition, each intracellular pathway does not operate in isolation; the pathways interact through mechanisms such as cross-talk [85] or scaffolding proteins [86]. Such interactions between the various pathways allow cells to fine-tune an appropriate response to a wide range of external stimuli. An example is the interaction between the P13/AKT/mTOR and MAPK pathways, both of which have key roles in the regulation of cell growth, proliferation, and survival. There are several points at which these pathways interact, facilitating coordinated control of downstream targets; thus, following activation the MAPK pathway can phosphorylate and activate components of the PI3K/AKT/mTOR pathway, amplifying the signalling output [87].

The intracellular signalling pathways described in this article have a number of features in common. Firstly, they provide a mechanism for altered gene expression in response to extracellular stimuli. Secondly, these pathways are involved (although by differing mechanisms) in cell growth, differentiation and survival, immune response, and inflammation. Thirdly, dysregulation of these pathways has been implicated in the pathogenesis of a range of disorders; particularly in cancer, but also in disorders as diverse as neurodegenerative disease and osteoporosis. Fourthly, there is evidence that CoQ10 can promote the normal functioning of all of these pathways. In particular, there is considerable evidence from preclinical studies for the role of CoQ10 in the activation of the Nrf2 pathway and inhibition of the NF-κB pathway, as summarised in Table 1 and Table 2, respectively. Dysregulation of intracellular signalling pathways can occur for a number of reasons, including mitochondrial dysfunction and oxidative stress. Given the role of CoQ10 in promoting normal mitochondrial function and as an antioxidant, it follows that CoQ10 has the potential to ameliorate the conditions leading to pathway dysfunction; in addition, there is evidence that CoQ10 can beneficially affect the action of various components in each of these pathways.

Dysregulation of signalling pathways can have the following clinical consequences: (i) Dysregulation of pathways involved in cell growth, proliferation, and apoptosis can result in uncontrolled cell growth and tumour formation. (ii) Signalling pathways are essential for immune cell activation, differentiation, and function; pathway dysregulation can result in immune deficiency and increased risk of infection, as well as autoimmune disease. (iii) Signalling pathways are involved in neurotransmitter release, receptor function, and neuronal communication; pathway dysfunction can result in neurological disorders. (iv) Signalling pathways regulate metabolism, including glucose uptake and lipid metabolism; pathway dysregulation can result in metabolic disorders like diabetes and obesity. (v) Dysfunction of signalling pathways involved in regulating blood pressure, heart contractility, and blood vessel function can contribute to the development of cardiovascular disease. The clinical consequences of intracellular signalling pathway dysfunction has been summarised in Table 3.

As noted above, many of the intracellular signalling pathways described in this article are involved in the development of cancer; in this regard the potential role of supplemental CoQ10 in the prevention or treatment of cancer has been reviewed by Mantle et al. [88]. It is also of note that CoQ10 interacts with peroxisome proliferator-activated receptor (PPAR)-related pathways, in the regulation of mitochondrial function, oxidative stress, inflammation, and lipid metabolism. CoQ10 first activates the AMPK pathway, resulting in increased expression of PPAR$\alpha$ and PPAR$\gamma$, leading to improved mitochondrial function, improved lipid metabolism, and reduced inflammation [89].

**Table 3 ijms-26-11024-t003:** Clinical consequences of intracellular signalling pathway dysfunction.

Pathway	Cellular Function of Pathway	Principal Clinical Consequence of Pathway Dysfunction
Nrf2/NQO1/	Protection against oxidative stress	Neurodegenerative disorders, including Parkinson’s disease, Alzheimer’s disease, multiple sclerosis, and amyotrophic lateral sclerosis [90]
Nf-κB	Regulation of immune response and inflammation	Autoimmune disorders, including rheumatoid arthritis, lupus erythematous, and IBS [91]
P13K/AKT/mTOR	Regulation of cell growth and proliferation	Cancer, including breast, colon, and skin cancers [92]
MAPK	Control of cell growth, differentiation and survival	Cancer, including colon, pancreatic, lung, and skin cancers [93]
JAK/STAT	Regulation of immune response and inflammation; cell growth and proliferation	Autoimmune disorders (e.g., rheumatoid arthritis, IBS) and cancer (e.g., haematological cancers, breast and lung cancer) [94]
Wnt/β-catenin	Normal embryonic development; control of cell growth	Embryonic abnormalities; cancer [95]
AMPK/YAP/OPA1	Control of energy metabolism	Cardiovascular disease; metabolic disorders (diabetes, obesity) [96]
Hedgehog	Normal embryonic development	Developmental disorders (e.g., craniofacial defects); cancer (e.g., basal cell carcinoma) [97]

As noted in Table 3, dysregulation of the Nrf2 pathway is principally associated with neurodegenerative disorders; reduced activity of the Nrf2 pathway is associated with Alzheimer’s disease [98], Parkinson’s disease [99], amyotrophic lateral sclerosis [100], Huntington’s disease [101], and Friedreich’s ataxia [102]. However, it is of note that in addition to the principal disorders listed in Table 3, most of the pathways have been implicated in a variety of other disorders. As an example, in addition to neurodegenerative disorders associated with reduced activity of the Nrf2/NQO1 pathway, other disorders that have been linked with the reduced activity of this pathway include cardiovascular disease [103], hepatic disease [104], renal disease [105], pulmonary disease [106], gastrointestinal disease [107], and musculoskeletal disease [108].

Intracellular signalling pathways are therapeutic targets in a variety of disorders. Thus some types of cancer therapy may target intracellular pathways that drive uncontrolled cell growth: examples include inhibitors of the MAPK or P13/AKT/mTOR pathways, such as trametinib for melanoma and everolimus for breast cancer [93,109]. Further examples of medicines, either approved or undergoing clinical trials, targeting intracellular signalling pathways have been summarised in Table 4.

It is also of note that the relevance of intracellular signalling pathways in clinical chemistry has become increasingly recognised, with components of the various pathways acting as potential biomarkers for the diagnosis and monitoring of a number of diseases. Examples of intracellular signalling pathway components used as clinical biomarkers include RAS proteins for the early detection of cancer and SMAD proteins for the diagnosis of immune disorders such as multiple sclerosis and Crohn’s disease [131,132]. However, in general terms, the use of intracellular signalling pathway components as clinical biomarkers for the diagnosis or prognosis of disease remains an area still in the relatively early stages of research. 

An area of relevance to intracellular signalling pathways is hormesis—a process in which a low dose of a stressor, which would be harmful at higher doses, triggers a beneficial response within the body. Hormesis promotes human health by activating the body’s defence and repair mechanisms, and hormesis activated by drugs or natural compounds is emerging as a promising preventive and therapeutic strategy in many chronic diseases [133,134]. An example of hormesis is the activation of the Nrf2 pathway, resulting in upregulation of cytoprotective antioxidant enzymes. In this regard, CoQ10 could be considered as a hormetic drug as it is able to induce health benefits at moderate doses (typically of the order of 100–200mg/day in clinical studies) by activating antioxidant pathways (Nrf2/NQO1) or inhibiting inflammatory pathways (NF-κB) in order to prevent or attenuate detrimental consequences of oxidative stress and chronic inflammation. However, it is of note that very high doses (up to 3000 mg/day) of CoQ10 used in some clinical studies reported no significant adverse effects [135].

In summary, in this article the importance of intracellular signalling pathways in the pathogenesis of a wide range of disorders has been highlighted. Supplementation with CoQ10 has been shown to be of benefit in a number of such disorders [88,136,137,138,139], although the mechanisms involved are not completely understood. In this article, the evidence we have reviewed provides a supporting rationale that the beneficial role of CoQ10 in these disorders occurs via mediation of intracellular signalling pathways, as summarised in Figure 1. Intracellular signalling pathways have primary effects in the form of first-line responses, such as the activation of protein kinases, and secondary effects such as the switching on or off of genes, or changes in cell metabolism. It is apparent from the data listed in Table 1 and Table 2 that supplementary CoQ10 has both primary and secondary effects on intracellular signalling pathways. A limitation of the present article is that most of the studies reviewed were preclinical; relatively few clinical studies have been carried out to specifically investigate the mediation of intracellular signalling pathways by CoQ10. As will be noted from Table 4, there is a precedent for developing novel medicines targeting intracellular signalling pathways, such as CoQ10, and further research into the clinical applications of the latter is now warranted.

## Figures and Tables

**Figure 1 ijms-26-11024-f001:**
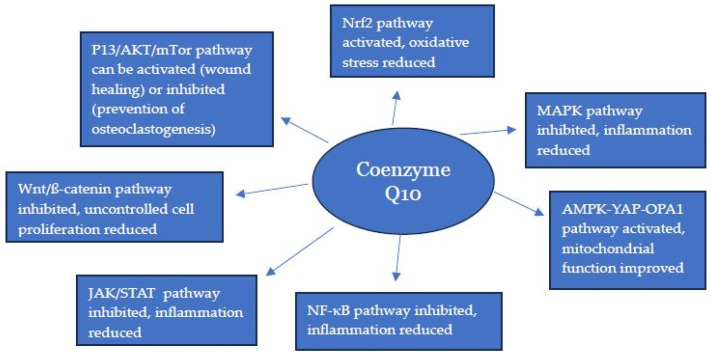
Effect of coenzyme Q10 on cellular signalling pathways.

**Table 1 ijms-26-11024-t001:** Summary of studies investigating the effect of CoQ10 on the Nrf2 pathway.

Outcome of CoQ10 Supplementation	System Studied	Study
Nrf2 increased, antioxidant enzymes increased, fibrosis decreased.	Fibrogenesis in mouse liver	Choi et al. [16]
Nrf2 increased, inflammation decreased.	Exercise training in rats	Pala et al. [17]
Nrf2 increased, oxidative stress reduced, inflammation reduced.	Liver inflammation in mice	Yang et al. [18]
Nrf2 increased, antioxidant enzymes increased, cell viability increased.	Neurotoxicity in PC12 cells	Li et al. [19]
Nrf2 increased, antioxidant enzymes increased, improved renal function.	Nephrotoxicity in mice	Kabel & Elkhoely [20]
Nrf2 increased, decreased oxidative stress, inflammation, and apoptosis.	Ulcerative colitis in rats	Khodir et al. [21]
Nrf2 increased, oxidative stress decreased.	Diabetes in rats	Samimi et al. [22]
Nrf2 increased, oxidative stress and apoptosis decreased.	Spinal cord injury in rats	Li et al. [15]
Nrf2 increased, oxidative stress, inflammation, and apoptosis decreased.	Lead induced neurotoxicity in rats	Yousef et al. [23]
Nrf2 increased, oxidative stress, and apoptosis decreased, improved cell viability.	Cisplatin induced toxicity in rat cardiomyocytes	Zhao [24]
Nrf2 increased, oxidative stress decreased.	Diabetic nephropathy in mice	Sun et al. [25]
Nrf2 increased, oxidative stress, inflammation, and apoptosis reduced.	Hepatic ischaemia–reperfusion injury in rats	Mahmoud et al. [26]
Nrf2 increased, oxidative stress and inflammation reduced, decreased blood pressure.	Pre-eclampsia in rats	Li et al. [27]
Nrf2 increased, oxidative stress, inflammation and apoptosis reduced.	Lead induced nephrotoxicity in rats	Al-Megrin et al. [28]
Nrf2 increased, oxidative stress, inflammation, and apoptosis reduced, improved liver function.	Thioacetamide induced liver toxicity in rats	Hussein et al. [29]
Nrf2 increased, oxidative stress, inflammation, and apoptosis reduced, allergy status improved.	Allergic rhinitis/asthma in mice	Du et al. [30]
Nrf2 increased, oxidative stress and apoptosis decreased, aneurysm formation reduced.	Intracranial aneurysm in mice	Huang et al. [31]
Nrf2 increased, oxidative stress and inflammation reduced.	Vincristine induced peripheral neuropathy in rats	Elsamy et al. [32]
Nrf2 increased, oxidative stress reduced.	Arsenic/chromium neurotoxicity in mice	Tripathy et al. [33]
Nrf2 increased, oxidative stress and inflammation decreased, cognition improved.	Chemotherapy induced cognitive impairment in mice	Kaur et al. [34]
Nrf2 increased, oxidative stress decreased, loss of retinal cells reduced.	Porcine retinal explant degeneration	Deppe et al. [35]
Nrf2 increased, oxidative stress decreased.	Diabetes in rats	Samimi et al. [13]
Nrf2 increased, oxidative stress, inflammation, and apoptosis reduced.	Testicular damage in rats	Arafa et al. [36]
Nrf2 increased, oxidative stress and apoptosis decreased.	Arsenic/chromium hepatotoxicity in mice	Tripathi et al. [37]

**Table 2 ijms-26-11024-t002:** Summary of studies investigating the effect of CoQ10 on the NF-κB pathway.

Outcome of CoQ10 Supplementation	System Studied	Study
NF-κB downregulated, oxidative stress, inflammation and apoptosis reduced	Mouse model of Parkinson’s disease	Kooncumchoo et al. [41]
NF-κB downregulated, oxidative stress, inflammation and apoptosis reduced	Acetaminophen induced liver toxicity in rats	Fouad et al. [42]
NF-κB downregulated, oxidative stress reduced	Neuropathic pain in diabetic mice	Zhang et al. [43]
NF-κB downregulated, oxidative stress and inflammation reduced	Trichloroacetic acid induced hepatocellular carcinoma in rats	Fouad et al. [44]
NF-κB downregulated, inflammation reduced	Wound healing in rats	Yoneda et al. [45]
NF-κB downregulated, inflammation reduced	Oxazolone induced dermatitis in mice	Li et al. [46]
NF-κB downregulated, inflammation reduced	Amyloid induced inflammation in PC12 cells	Li et al. [47]
NF-κB downregulated, inflammation reduced	Ischaemia–reperfusion injury in skeletal muscle of rats	Boroujeni et al. [48]
NF-κB downregulated, oxidative stress, inflammation and apoptosis reduced	Restraint induced depression in mice	Salehpour et al. [49]
NF-κB downregulated, oxidative stress and inflammation reduced	Cerebral malaria in mice	Nyariki et al. [50]
NF-κB downregulated, oxidative stress and inflammation reduced	Anthracycline induced toxicity in human cardiomyocytes	Quagliariello et al. [51]
NF-κB downregulated, mitochondrial function improved, inflammation reduced	Lipopolysaccharide induced lung injury in rats	Ali et al. [52]
NF-κB downregulated, oxidative stress and inflammation reduced	Radiation induced enteropathy in rats	Mohamed & Said [53]
NF-κB downregulated, oxidative stress, inflammation and apoptosis reduced	Propionic acid induced cerebral injury in rats	Alhusaini et al. [54]
NF-κB downregulated, oxidative stress and inflammation reduced	Cadmium/titanium induced liver toxicity in rats	Abd-Elkahim et al. [55]
NF-κB downregulated, oxidative stress and inflammation reduced	Cerebral ischaemia–reperfusion injury in rats	Fakharaldeen et al. [56]
NF-κB downregulated, tumour cell invasiveness decreased	Glioblastoma in mice	Frontinan-Rubio et al. [57]
NF-κB downregulated, oxidative stress and inflammation reduced	Doxorubicin induced liver toxicity in rats	Mansour et al. [58]
NF-κB downregulated, oxidative stress and inflammation reduced	Cadmium cardiotoxicity in rats	Antar et al. [59]
NF-κB downregulated, inflammation and apoptosis reduced	Intracerebral haemorrhage in mice	Yang et al. [60]

**Table 4 ijms-26-11024-t004:** Medicines targeting intracellular signalling pathways.

Pathway	Medicine	Disorder	Reference
Nrf2/NQO1	Nrf2 activators: omaveloxolone dimethyl fumarate	Friedreich’s ataxia multiple sclerosis	Lynch et al. [110] Okuda et al. [111]
NF-κB	NF-κB inhibitors: bortezomib vorinostat	myeloma cutaneous T-cell lymphoma	Cengiz-Seval et al. [112] Kavanaugh et al. [113]
P13K/AKT/mTOR	P13K inhibitors: alpelisib copanlisib AKT inhibitors: capivasertib ipatasertib mTOR inhibitors: everolimus temsirolimus	breast cancer endometrial cancer breast, prostate cancer prostate cancer renal cell cancer renal cell cancer	Copur [114] Santin et al. [115] Turner et al. [116] Sutaria et al. [117] Amato [118] Goudarzi et al. [119]
MAPK	RAF inhibitors: vemurafenib dabrafenib MEK inhibitors: trametinib selumetinib	melanoma melanoma melanoma neurofibroma	Aires-Lopez [120] Long et al. [121] Thota et al. [122] Kim et al. [123]
JAK/STAT	JAK inhibitors: baricitinib upadacitinib STAT inhibitors: OBP-31121	rheumatoid arthritis atopic dermatitis various cancers	Urits et al. [124] Simpson et al. [125] Oh et al. [126]
AMPK/YAP/OPA1	AMPK activators: metformin phenformin	type II diabetes cancer	Aguilar-Recarte [127] Zhang et al. [128]
Hedgehog	SMO inhibitors: vismodegib glasdegib	basal cell carcinoma leukaemia	Sekulic et al. [129] Fersing [130]

## Data Availability

No new data were created or analysed in this study. Data sharing is not applicable to this article.

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
