# Peer review of "Coenzyme Q10 and Intracellular Signalling Pathways: Clinical Relevance"

_ijms, 2025, doi:10.3390/ijms262211024_

Round 1

Reviewer 1 Report

Comments and Suggestions for Authors

Mantle’s review regarding clinical relevance of Coenzyme Q10 and intracellular, covers a highly relevant and intriguing topic, considering the great and growing interest in CoQ in health and disease, as well as its widespread use as a dietary supplement. Although the mechanisms of CoQ within the cell are mostly known for its key role in the electron transport chain, its functions beyond mitochondria and more broadly in organismal physiology remain largely undefined. In particular, its influence on various pathways regulating metabolism, antioxidant responses, and signaling mechanisms are unclear.

In this review, the author attempts to summarize the role of CoQ in different signaling pathways, linking to different diseases and current pharmacological approaches. THe manuscript provides a comprehensive overview of current studies on the implications of CoQ across various pathways and discusses pharmacological targeting of these mechanisms. However, several issues need to be addressed to strengthen the review. I strongly encourage the author to address the listed issues to make the review and its take-home message clearer, as well as to make the topic more transparent to an audience that may be naive to CoQ biology and its mechanism beyond ETC, and/or to the physiological relevance of the signalling pathways described.

  • In most sections, no references are provided to the original studies or reviews that first identified or described the signalling pathways mentioned throughout the manuscript. The author should include appropriate citations. For this purpose, review articles are fine too. Moreover, it is not always clear what are the physiological as well as cellular downstream or general mechanisms of these pathways. It should be clear to the reader, in the first place, why the cell activates them in certain conditions and why and how CoQ could play a role in that context. Adding these details, even briefly, could allow the reader to better understand the context of the pathways described.
  • Many of the pathways described (for example, PI3K/AKT/mTOR) are known to be activated in various cancer types, where they contribute to improved, for instance, antioxidant defenses or metabolic reprogramming toward glycolysis. The author did not always mention this aspect.
  • In relation to the previous comments, given that most of the signalling pathways described are pivotal in cancer growth or proliferation or cell death escape (like for instance boosting antioxidant production through Keap1/Nrf2/ARE signaling pathway or some other pathways described in this review), it would be valuable for the author to comment on the dynamics of CoQ in cancer, particularly as an antioxidant, given the controversial role of antioxidants in tumor progression and therapy resistance.
  • The Wnt/β-catenin pathway plays a fundamental role in cardiac regeneration and tissue repair (see, for instance, PMID: 39725339). Given the widespread use of CoQ as a nutraceutical supplement, especially in cardiovascular disease settings, the section discussing this pathway should include relevant observations or comments linking CoQ to cardiac physiology and regeneration relevant signalling pathways.
  • Some studies, including work in zebrafish models (PMID: 30171034), have reported that CoQ10 and its more soluble analogue, idebenone, can act as partial agonists of peroxisome proliferator-activated receptors α (PPARα) and γ (PPARγ). Since these receptors regulate lipid metabolism and inflammation, which are druggable key processes in several metabolic disorders, including a discussion on the potential involvement of CoQ in PPAR-related pathways would greatly strengthen the review.
  • Overall, understanding CoQ interactions with the different pathways described could provide a stronger foundation and greater plausibility for the use of CoQ as a nutraceutical or adjuvant therapy across a wide range of diseases, from cancer and obesity to diabetes and statin-induced myopathies, with the potential for extending its application far beyond CoQ deficiency-related disorders. Most of the pathways described are closely linked to cellular metabolism, and given the widespread physiological roles of CoQ alongside the limited understanding of its mechanisms/ functions outside mitochondria, it remains unclear whether CoQ directly modulates these pathways or if its effects are secondary to altered cellular bioenergetics/ or enhanced antioxidant defenses. Clarifying this distinction throughout the manuscript would be important. It would highlight the current limitations and gaps in understanding the mechanisms underlying the complex and multifaceted interactions of CoQ within intracellular signaling pathways.

Author Response

Mantle’s review regarding clinical relevance of Coenzyme Q10 and intracellular, covers a highly relevant and intriguing topic, considering the great and growing interest in CoQ in health and disease, as well as its widespread use as a dietary supplement. Although the mechanisms of CoQ within the cell are mostly known for its key role in the electron transport chain, its functions beyond mitochondria and more broadly in organismal physiology remain largely undefined. In particular, its influence on various pathways regulating metabolism, antioxidant responses, and signaling mechanisms are unclear.

In this review, the author attempts to summarize the role of CoQ in different signaling pathways, linking to different diseases and current pharmacological approaches. The manuscript provides a comprehensive overview of current studies on the implications of CoQ across various pathways and discusses pharmacological targeting of these mechanisms. However, several issues need to be addressed to strengthen the review. I strongly encourage the author to address the listed issues to make the review and its take-home message clearer, as well as to make the topic more transparent to an audience that may be naive to CoQ biology and its mechanism beyond ETC, and/or to the physiological relevance of the signalling pathways described.

In most sections, no references are provided to the original studies or reviews that first identified or described the signalling pathways mentioned throughout the manuscript. The author should include appropriate citations. For this purpose, review articles are fine too.

Additional references have now been provided.

Moreover, it is not always clear what are the physiological as well as cellular downstream or general mechanisms of these pathways. It should be clear to the reader, in the first place, why the cell activates them in certain conditions and why and how CoQ could play a role in that context. Adding these details, even briefly, could allow the reader to better understand the context of the pathways described.

An explanatory note has been added in the Introduction.

Many of the pathways described (for example, PI3K/AKT/mTOR) are known to be activated in various cancer types, where they contribute to improved, for instance, antioxidant defenses or metabolic reprogramming toward glycolysis. The author did not always mention this aspect. A quick search for “cancer” shows 45 occurrences in the paper, of which 17 are in the references- see also next item.

In relation to the previous comments, given that most of the signalling pathways described are pivotal in cancer growth or proliferation or cell death escape (like for instance boosting antioxidant production through Keap1/Nrf2/ARE signaling pathway or some other pathways described in this review), it would be valuable for the author to comment on the dynamics of CoQ in cancer, particularly as an antioxidant, given the controversial role of antioxidants in tumor progression and therapy resistance.

Reference to the role of CoQ10 in the prevention or treatment of cancer has been made in the Discussion section.

The Wnt/β-catenin pathway plays a fundamental role in cardiac regeneration and tissue repair (see, for instance, PMID: 39725339). Given the widespread use of CoQ as a nutraceutical supplement, especially in cardiovascular disease settings, the section discussing this pathway should include relevant observations or comments linking CoQ to cardiac physiology and regeneration relevant signalling pathways.

A note covering the above has been added to section 7.

Some studies, including work in zebrafish models (PMID: 30171034), have reported that CoQ10 and its more soluble analogue, idebenone, can act as partial agonists of peroxisome proliferator-activated receptors α (PPARα) and γ (PPARγ). Since these receptors regulate lipid metabolism and inflammation, which are druggable key processes in several metabolic disorders, including a discussion on the potential involvement of CoQ in PPAR-related pathways would greatly strengthen the review.

A note re the above has been included in the Discussion.

Overall, understanding CoQ interactions with the different pathways described could provide a stronger foundation and greater plausibility for the use of CoQ as a nutraceutical or adjuvant therapy across a wide range of diseases, from cancer and obesity to diabetes and statin-induced myopathies, with the potential for extending its application far beyond CoQ deficiency-related disorders. Most of the pathways described are closely linked to cellular metabolism, and given the widespread physiological roles of CoQ alongside the limited understanding of its mechanisms/ functions outside mitochondria, it remains unclear whether CoQ directly modulates these pathways or if its effects are secondary to altered cellular bioenergetics/ or enhanced antioxidant defenses.

A note re primary/secondary effects of Q10 on signaling pathways has been included in the Discussion.

Clarifying this distinction throughout the manuscript would be important. It would highlight the current limitations and gaps in understanding the mechanisms underlying the complex and multifaceted interactions of CoQ within intracellular signaling pathways.

Reviewer 2 Report

Comments and Suggestions for Authors

This review explores the effects of CoQ10 on major intracellular signaling pathways, including the Nrf2/NQO1, NF-kB, PI3K/AKT/mTOR, MAPK, JAK/STAT, WNT/β-catenin, AMPK-YAP-OPA1, and Hedgehog (Hh) pathways. However, the current version is not suitable for publication, primarily due to the following issues:

1, Throughout the background section, as well as the first two paragraphs of sections such as "2. The Nrf2/NQO1 pathway," "6. The JAK/STAT pathway," and "7. The Wnt/β-catenin pathway," no references are cited. This is highly unusual and substandard for a review article.

2, Some sections, such as "6. The JAK/STAT pathway," "7. The Wnt/β-catenin pathway," "8. The AMPK-YAP-OPA1 pathway," and "9. The Hedgehog pathway," contain only 1-2 references in their main text, which is insufficient to support the related descriptions and makes the conclusions lack persuasiveness.

3, Under what pathological conditions is CoQ10 supplementation necessary, and what are the effects of such supplementation? The author did not provide this background information and directly delved into the descriptions of intracellular signaling pathways, which is inappropriate.

4, The effects of CoQ10 are closely linked to mitochondrial function, and mitochondria can be considered a core component in this context. However, why does the author rarely mention anything related to mitochondria?

5, Regarding the impact of CoQ10 on the intracellular signaling pathways listed in this manuscript, is there a distinction between primary and secondary effects?

6, The inclusion of appropriate figures would help more clearly illustrate how CoQ10 functions through major intracellular signaling pathways.

7, The abstract lacks information about the conclusions of this review and its significance.

Author Response

This review explores the effects of CoQ10 on major intracellular signaling pathways, including the Nrf2/NQO1, NF-kB, PI3K/AKT/mTOR, MAPK, JAK/STAT, WNT/β-catenin, AMPK-YAP-OPA1, and Hedgehog (Hh) pathways. However, the current version is not suitable for publication, primarily due to the following issues:

1, Throughout the background section, as well as the first two paragraphs of sections such as "2. The Nrf2/NQO1 pathway," "6. The JAK/STAT pathway," and "7. The Wnt/β-catenin pathway," no references are cited. This is highly unusual and substandard for a review article.

Additional references have now been added.

2, Some sections, such as "6. The JAK/STAT pathway," "7. The Wnt/β-catenin pathway," "8. The AMPK-YAP-OPA1 pathway," and "9. The Hedgehog pathway," contain only 1-2 references in their main text, which is insufficient to support the related descriptions and makes the conclusions lack persuasiveness.

Additional references have now been added.

3, Under what pathological conditions is CoQ10 supplementation necessary, and what are the effects of such supplementation? The author did not provide this background information and directly delved into the descriptions of intracellular signaling pathways, which is inappropriate.

A section has been added in the Introduction (second paragraph) to cover this point.

4, The effects of CoQ10 are closely linked to mitochondrial function, and mitochondria can be considered a core component in this context. However, why does the author rarely mention anything related to mitochondria?

Mitochondria are discussed throughout the paper, e.g., section 2 on page 2, Table 2 on page 4, section 8 on page 7, and section 10 (Discussion) on page 8. However, the emphasis in this review is more on the effects of oxidative stress and chronic inflammation on signaling pathways than it is on the effects of mitochondrial dysfunction.

5, Regarding the impact of CoQ10 on the intracellular signaling pathways listed in this manuscript, is there a distinction between primary and secondary effects?

A note has been added in the Discussion to cover this point.

6, The inclusion of appropriate figures would help more clearly illustrate how CoQ10 functions through major intracellular signaling pathways.

As the paper includes 4 tables and an extensive reference list, the article is already rather long to include additional figures, particularly as each of the 8 pathways described would require its own figure.

7, The abstract lacks information about the conclusions of this review and its significance.

The abstract has been re-drafted with regard to the above points.

Round 2

Reviewer 2 Report

Comments and Suggestions for Authors

The revised manuscript could be accepted.